# High-Risk Acute Myeloid Leukemia: A Pediatric Prospective

**DOI:** 10.3390/biomedicines10061405

**Published:** 2022-06-14

**Authors:** Fabiana Cacace, Rossella Iula, Danilo De Novellis, Valeria Caprioli, Maria Rosaria D’Amico, Giuseppina De Simone, Rosanna Cuccurullo, William G. Wierda, Kris Michael Mahadeo, Giuseppe Menna, Francesco Paolo Tambaro

**Affiliations:** 1Unità Operativa di Trapianto di Cellule Staminali Ematopoietiche e Terapie Cellulari, Azienda Ospedaliera di Rilievo Nazionale Santobono-Pausilipon, 80123 Napoli, Italy; fabianacacace87@gmail.com (F.C.); v.caprioli@santobonopausilipon.it (V.C.); m.damico@santobonopausilipon.it (M.R.D.); g.desimone@santobonopausilipon.it (G.D.S.); 2Dipartimento di Medicina Clinica e Chirurgia, Azienda Ospedaliera Universitaria Federico II, 80100 Napoli, Italy; rossella.iula@gmail.com; 3Centro di Ematologia e Trapianto, Ospedale San Giovanni di Dio e Ruggi D’Aragona, 84131 Salerno, Italy; danidenov90@gmail.com; 4Unità Operativa di Ematologia, Azienda Ospedaliera di Rilievo Nazionale Santobono-Pausilipon, 80123 Napoli, Italy; r.cuccurullo@santobonopausilipon.it (R.C.); g.menna@santobonopausilipon.it (G.M.); 5Department of Leukemia, University of Texas MD Anderson Cancer Center, Houston, TX 77030, USA; wwierda@mdanderson.org; 6Pediatric Stem Cell Transplantation and Cellular Therapy, CARTOX Program, University of Texas at MD Anderson Cancer Center, Houston, TX 77030, USA; kmmahadeo@mdanderson.org

**Keywords:** AML, pediatric, risk stratification, outcome, target therapy

## Abstract

Pediatric acute myeloid leukemia is a clonal disorder characterized by malignant transformation of the hematopoietic stem cell. The incidence and the outcome remain inferior when compared to pediatric ALL, although prognosis has improved in the last decades, with 80% overall survival rate reported in some studies. The standard therapeutic approach is a combined cytarabine and anthracycline-based regimen followed by consolidation with allogeneic stem cell transplantation (allo-SCT) for high-risk AML and allo-SCT for non-high-risk patients only in second complete remission after relapse. In the last decade, several drugs have been used in clinical trials to improve outcomes in pediatric AML treatment.

## 1. Introduction

Pediatric acute myeloid leukemia (AML) is a heterogeneous clonal disorder characterized by malignant transformation of the hematopoietic stem cell (HSC). Pediatric AML is less common than acute lymphoblastic leukemia (ALL); while ALL accounts for 80% of all pediatric acute leukemias, AML barely reaches 15–20%. Pediatric AML has a bimodal age distribution, with higher incidence in patients younger than 2 years and a second peak in adolescents up to 10 to 20 years old [1].

Pediatric AML is generally characterized by poorer prognosis compared to ALL although outcomes have improved in the recent years. The 3-year overall survival (OS) is 70% or even higher [2,3,4]. Notably, 30% of children with AML relapse after treatment with very poor OS. The AML BFM study group reported a 5-year OS of 45%± 4%, with fewer early deaths (8.1%vs.2.2%) [5,6].

While our current understanding of pediatric AML may be influenced by adult studies, substantial differences in the mutational landscape and tolerance to treatment have emerged between children and adults [7].The real challenge in the treatment of pediatric AML is achieving durable complete remission (CR).

Understanding the biology of pediatric-AML-specific gene mutations may enable new and more specific up-front treatment approaches in order to increase the rates and the duration of the remission and to guarantee more therapeutic options in case of relapse. The aim of our review is to analyze the efficacy and safety of new therapeutic agents already approved for adult AML and translate them for application to pediatric high-risk AML, focusing on molecular pathways and potential therapeutic indications.

## 2. Pathophysiology of Pediatric AML

While the majority of pediatric AML occurs de novo, there is an increased incidence of AML reported in certain hereditary disorders such as Fanconi anemia, Kostmann syndrome, Shwachman–Diamond syndrome, and Diamond–Blackfan anemia, which may be related to alterations of DNA repair or detoxification of reactive oxygen species genes to pathogenetic variants in telomere biology genes or ribosome function [8].Children with Down syndrome (DS) have an increased risk of developing AML, particularly the megakaryoblast (MK) subtype. Indeed, the most common genetic factor associated with the development of AML is trisomy 21 [9].Germ-line mutations were found in families with a high risk of AML, suggesting a familial predisposition [10].DS patients also harbor a GATA1 mutation that, as broadly established, can lead to the development of a transient myeloproliferative disease (TDM), which commonly resolves without any treatment. Residual cells may undergo apoptosis or acquire additional mutations leading to acute megakaryocytic leukemia (AMKL) with an average latency of 3 years. Nevertheless, these patients typically have less-aggressive disease, with a significantly longer disease-free survival (DFS) compared to other types of pediatric AML. This is possibly due the fact that DS-AMKL requires less intense therapy to achieve a cure compared to non-DS AMKL, showing that children with AML-DS are more responsive to chemotherapy [11,12].

AML is derived from a clonal proliferation and the abnormal differentiation of myeloid stem cells as a consequence of two sequential genetic events. This process is known as the two-hit model of leukemogenesis [13]. In AML, there are two different types of relevant mutations: class I mutations frequently result in uncontrolled proliferation and include receptor or cytoplasmatic–nuclear tyrosine kinase mutations such as FLT3, K/NRAS, TP53, and c-KIT, in 20–25%, 40%, 2%, and 12% cases, respectively, and do not affect differentiation. In contrast, class II mutations involve cellular differentiation arrest and/or self-renewal [7,14]. The most common type II cytogenetic abnormalities in children are t(8;21)(q22;q22), involving core-binding factor (CBF) β AML, as well as inv 16 or t(16;16), and t(15;17)(q22;q21) in acute promyelocytic leukemia (APL). Specific pediatric translocations rarely detected in adults include t(1;22)(p13;q13) and t(11;12)(p15;p13). The NPM1 and CEBPA, class II mutations, are observed in 5–10% and 4% of cases, respectively, and they are associated with a better prognosis [15,16]. Although attractive, this model represents an oversimplification of AML pathogenesis; many approaches revealed the presence of novel mutations and absence of tyrosine kinase lesions, particularly in normal-karyotype AML, reducing the relevance of the two-hit model [17].

## 3. Therapeutic Considerations: Past and Future

The WHO classification for hematological malignancies, revised in 2016, integrated genetic characteristics, such as karyotypes and molecular aberrations, with morphology, immunophenotype, and clinical presentation, but has limited application in children, since cytogenetic and genetic abnormalities are uncommon as compared to adult AML [18]. As such, pediatric AML is classified as “not-otherwise-specified” [19].

New discoveries in AML genetic alterations were applied to pediatric patients, improving risk stratification and therapeutic approaches. The TARGET project, an analysis of the molecular aberrations in pediatric AML, showed that the mutation rate is lower in pediatric than in adult AML, with different somatic aberrations including structural changes, aberrant DNA methylation, and specific pediatric mutations [20]. In particular, RAS, KIT, and FLT3 class I mutation types are the most common mutated genes while DNMT3, IDH1, and IDH2 gene mutations are rare [21].

Fusion genes specifically identified in pediatric AML and associated with a grim prognosis include CBFA2T3-GLS2 and NUP98-NSD1.CBFA2T3-GLS2 is a chimeric transcript derived from a cryptic inversion of the telomeric region of chromosome 16 and expressed especially in non-DS FAB M7 AML. This gene mutation is present in patients younger than 5 years old and is characterized by high bone marrow blast count and extra-medullary involvement [20]. In contrast, NUP98-NSD1 can be found in 3.8% cases of pediatric AML and is the most frequent NUP98 rearrangement. It is usually associated with other chromosomal abnormalities, particularly trisomy 8, and other genetic mutations, such as FLT3-ITD, WT1, and CEBPA, and appears to play a role in histone methylation and acetylation [22].

Diagnostic approaches in pediatric AML may be influenced by experience in adults [7]. Treatment decisions are based on risk stratification and further guided by the initial response to treatment [5,6,23]. Most international pediatric study groups (AIEOP/BFM/FRANCE/UK/COG/Japan) define risk classification according to genetic/molecular abnormalities and response to treatment by the measurement of residual disease by flow and morphology [24]. In particular, favorable prognostic factors include t(8;21)(q22;q22)/RUNX1-RUNX1T1, t(15;17)(q22;q21)/PML-RARA, NPM1-mutated AML, and CEBPA double mutation [6].AML with MLL translocations has variable outcomes, depending on the associated translocation and occurs more frequently in children compared to in adults [25]. Chromosome 3q and 5q abnormalities and monosomy karyotype and high blast count at diagnosis are predictors of poor outcome [6,26].

FLT3 mutation is also associated with dismal prognosis in adult AML [27] and remains controversial in pediatrics [28,29,30]. However, a large meta-analysis of 10 trials including 1661 patients with pediatric AML showed a shorter OS for FLT3-ITD mutated AML [31]. AIEOP-BFM AML 2020 proposed risk stratification in three groups (standard, intermediate, and high-risk AML); AML is high risk when minimal residual disease (MRD) is ≥1% after induction course 1 or ≥0.1% at induction 2 or blast count is ≥5% at induction 1 (only if flow results are not informative) with one of genetic/molecular aberration at diagnosis, as described in Table 1 [24].

## 4. Novel Potential Therapies

Few changes were observed over the last forty years in the treatment algorithm of both pediatric and adult AML. The standard therapeutic approach is a combined cytarabine and anthracycline-based regimen followed by consolidation with allogeneic stem cell transplantation (allo-SCT) for high-risk AML and allo-SCT for non-high-risk patients only in second CR after relapse [32,33].In the last decade, several drugs such as epigenetic treatments, i.e., hypomethylating agents or histone-deacetylaseinhibitors, anti-CD33–ozogamicin conjugated monoclonal antibodies, and FLT3 and isocitrate dehydrogenase (IDH) inhibitors were used to treat pediatric and adult AML (Table 2).

### 4.1. Epigenetic Therapy

Physiological gene expression is regulated by epigenetic chromatin modifications, such as DNA methylation and histone acetylation [34,35]. Altered epigenetic regulation plays a role in pediatric AML [36,37]. Hypomethylating agents (HMAs) include two major compounds: 5-azacytidine (azacytidine) and 5-aza-2′-deoxycytidine (decitabine), which inhibit DNA methyl transferase (DNMT) and promote the switching on of tumor-suppressor genes and the downregulation of oncogenes and increased sensitivity to cytotoxic agents [38,39]. In particular, they have been clinically approved by the Food and Drug Administration (FDA) and the European Medicines Agency (EMA) for the treatment of myeloid malignancies in the elderly and in patients who are ineligible for more-intensive regimens [40,41,42,43].Two retrospective analyses conducted in children diagnosed with myelodysplastic syndrome (MDS) and relapsed/refractory (r/r) AML reported good clinical activity for both drugs, with an acceptable safety profile. In the first study, 24 children and young adults with MDS received azacytidine at the time of first diagnosis (16 patients) or relapse after allo-SCT (10 patients). The outcome with azacytidine as the first-line treatment in 16 pediatric MDS patients was analyzed: one patient obtained a complete clinical remission, and one showed complete marrow remission, while six experienced stable disease with hematologic improvement. Treatment discontinuation and dose reduction were necessary in one and in five children, respectively, due to impaired renal function and cytopenia [44].

The second retrospective study was are port of eight children and young adults with a median age of 4 years with r/r AML that evaluated the efficacy of low-dose decitabine. Three of eight patients reached a complete response, with the best responses observed after a median of 2.5 cycles (range 1–4 cycles). In this study, there were no severe side effects, except for neutropenia. An episode of febrile neutropenia occurred, but the overall infection rate was low [45]. Moreover, a multicenter, open-label, phase II study (NCT02450877) tested azacytidine in seven children and young-adult patients with AML in molecular relapse after first complete remission. Five patients had an MRD assessment, with four patients achieving either molecular stabilization or improvement, suggesting that azacytidine may reduce MRD prior to hematopoietic stem cell transplantation (HSCT) [46].

The first pediatric phase I study evaluating azacytidine combined with intensive fludarabine plus cytarabine-based chemotherapy(NCT01861002) included14 patients with r/r acute leukemia (AML *n* = 12; ALL *n* = 2) and showed promising results in terms of a high complete remission rate (CR: 7/12; 58%) [47].

A phase II trial in patients younger than 21 years with de novo AML to evaluate the eventual improvement in OS and event-free survival (EFS) (NCT03164057) through epigenetic priming with DNMT inhibitors prior to chemotherapy is currently ongoing. Decitabine was also investigated in a randomized two-arm phase II clinical trial (NCT01177540) comparing standard chemotherapy with or without decitabine in patients with AML aged1–16 years. Although there was a good tolerability profile, there was no difference in CR rate between the two arms [48].

Treatment in vitro with the histone deacetylation (HDAC) inhibitors, vorinostat and panobinostat, was associated with the activation of tumor-suppressor genes and cancer cell death [49]. In adult AML, HDAC inhibitors were evaluated alone and in combination with chemotherapy supporting a rationale for the use of HDAC inhibitors in combination with chemotherapy [50].

Interesting results were reported from a phase I trial (NCT02676323) evaluating the safety, pharmacokinetics, and pharmacodynamics of panobinostat in 17 pediatric patients with relapsed AML; 8 out of 17 patients (47%) achieved CR and 6 (75%) achieved undetectable MRD status [51].

The efficacy of vorinostat in pediatric relapsed AML is currently being evaluated in an ongoing clinical trial (NCT03263936); the results represent the first step in assessing of the utility of an HDAC inhibitor in pediatric patients.

Epigenetic treatment should be considered for a special group of pediatric AML patients carrying MLL translocations. The MLL fusion protein recruits a histone methyl transferase (HMT), DOT1L, resulting in the aberrant methylation of MLL gene targets and enhanced expression of leukemia-associated genes [52]. Since this aberration is more frequent in pediatric AML and is the most common in infant AML, there is a strong rationale for the development of inhibitors of this complex [53].

Preclinical studies demonstrated that the addition of pinometostat to conventional chemotherapy might sensitize MLL-associated leukemia cells to chemotherapy, by way of DOT1L inhibition. Nevertheless, a phase I trial of pinometostat, a small-molecule DOT1L inhibitor, conducted in children with r/r MLL-associated AML (NCT02141828) demonstrated only a transient reduction in leukemic blast counts [54]. A phase I/II trial of pinometostat plus standard chemotherapy (NCT03724084)in pediatric and adult patients with newly diagnosed MLL-rearranged AML is currently ongoing [55]. Moreover, results with combined pinometostat and FLT3 inhibitor were recently reported for pediatric AML patients, confirming that targeting DOT1L with pinometostat sensitizes AML cells to further treatment with the multi-kinase inhibitor sorafenib [56].

### 4.2. Immunotherapyand Immune Mediated-Chemotherapy

Immune-based therapy is a promising approach to overcome resistance to conventional chemotherapy in AML. Different types of immunotherapies are being evaluated in AML, including monoclonal antibodies, T-cell therapy, and cancer vaccines [57].

In pediatric AML, the most relevant immune-based therapy approach consists of targeting surface antigens, particularly CD33 (sialic-acid-binding immunoglobulin lectin, SIGLEC) and CD123 (IL3Rα), both highly expressed, although not exclusively, by AML cells [58].

Gemtuzumab ozogamicin (GO) is an anti-CD33 antibody conjugated to a cytotoxic agent, calicheamicin, with a potent antitumor effect against CD33-expressing cells [59]. Initial studies in adults demonstrated improved EFS and OS when combined with conventional chemotherapy for newly diagnosed CD33-positive AML in adults [60]. However, GO was removed from the market because of high rates of severe liver toxicity, including sinusoidal obstruction syndrome/veno-occlusive disease (SOS/VOD) [61]. Recently, lower doses of GO were shown to produce similar efficacy with less treatment-related morbidity and mortality [62,63]. This led the FDA in 2016 to reapprove GO in adult patients with de novo AML with favorable and intermediate cytogenetics or relapsed leukemia [64]. In 2018, the EMA reapproved GO combined with daunorubicin and cytarabine for the treatment of patients aged 15 years and above with previously untreated, de novo CD33-positive AML [65]. GO was subsequently studied in pediatric clinical trials as treatment for de novo and relapsed patients and demonstrated improvements in EFS and relapse rate when combined with chemotherapy in the favorable and intermediate-risk group [66,67]. In particular, patients in the GO-treated arm of the AAML 0531 trial showed 53.1% 3-year EFS (vs.46.9% in the non-GO-treated arm)with a significantly reduced (32.8% vs. 41.3%)risk of relapse at 3 years [68].

In adult high-risk AML patients, no benefit was observed in clinical trials combining GO with conventional therapy [69]; however, subgroups of pediatric patients with FLT3-ITD or those who expressed KMT2A rearrangements or high CD33 expression benefited from treatment with GO therapy [70,71]. Because of these observations, GO was approved by the FDA for children aged 2 years and older with relapsed AML [72]; similarly, the EMA approved GO for de novo AML patients aged 15 years and above. Since last year, based on the results of the AAML0531 trial, the FDA extended the indication of GO to newly diagnosed CD33-positive AML including pediatric patients aged 1 month and older.

CD123, the IL-3 receptor α-chain, is also expressed on the surface of AML blasts in the majority of cases [58]. CD123-targeted therapies remain in the early phases of research. Currently, several antibody conjugates are being studied, including CD123-targeting drug conjugates and a bi-specific molecule retargeting CD3-CD123 [73,74,75]. The normal physiological mechanisms to downregulate the immune system to reduce the risk of hyper-inflammation damage, in particular, immune checkpoint inhibitors, represents a novel treatment strategy for AML. The control of the immune response through the regulation of the checkpoints was recently tested in several cancers and was considered for the treatment of AML [76].The checkpoint inhibitors are deputed to control damages derived by hyperinflammation. Their inhibitors can induce cancer cell death, while also favoring an inflammatory response in AML cells avoiding immune surveillance [77].Checkpoint inhibition with PD-1/PD-L1 (e.g., nivolumab, pembrolizumab) or CTLA4 antibodies(e.g., ipilimumab) was evaluated in adult solid neoplasms, and AML is currently under investigation. Early results of phase I/II trials showed modest clinical efficacy for checkpoint inhibitors given as monotherapy; higher response rates were observed when combined with other agents such as HMAs [78,79].

Furthermore, PD-1 inhibition was also evaluated in adult patients with relapse after HSCT to restore donor chimerism [80].

Very few studies evaluating checkpoint inhibitors in pediatric AML are currently available; a phase I/II (NCT03825367) trial is currently evaluating the efficacy of nivolumab combined with azacytidine in pediatric r/r AML.

In contrast to lymphoid neoplasms, where different surface antigens such as CD19, CD22, and recently, CD20 and CD23 were targeted for cellular therapy chimeric antigen receptor T (CAR-T) cells leading to very good results, this treatment strategy has not yet been successfully applied to treatment for patients with AML [72]. The main reason for this has been the absence of universal myeloid antigens for therapeutic targeting. Moreover, multiple myeloid antigens are shared with the hematopoietic stem cell. As such, cell therapy could lead to the destruction of the stem compartment without “on target/off tumor” hematologic and non-hematologic toxicity [81]. However, CD33 and CD123 are both interesting target leukemia antigens, expressed by AML blasts in most cases. CAR-T cells targeting CD33 and CD123 showed potent antitumor activity in preclinical models [81,82].

Several phase I trials focused on the clinical evaluation of CD33 CAR-T cells in adults with r/r AML, and there have been a number of case reports and pilot studies showing the use of CAR-T cells in AML [72]. A phase I (NCT03126864) trial enrolling adult and pediatric patients with r/r AML, terminated in October 2019; it investigated the safety and tolerability of three different doses of autologous CAR-T cells, modified to express CD33-targetedCAR-T cells [75]. Moreover, other trials are currently ongoing withCD33-CAR-T cells for children (NCT03971799) and young adults (NCT03927261) with AML.

Trials evaluating the efficacy of anti-CD123 CAR-T are currently open or have recently been completed for patients with r/r AML. In pediatrics, two phase I trials (NCT02159495 and NCT04318678) are currently evaluating the efficacy of CD123-directed CAR-T cells in r/r AML patients older than 12 years up to 21 years of age [83].

### 4.3. Optimizing Chemotherapy: CPX-351

CPX-351, a liposomal combination of daunorubicin and cytarabine in a 5:1 molar ratio, is approved by the EMA for the treatment of adults with newly diagnosed, therapy-related AML (t-AML) or AML with MDS-related changes. The efficacy and safety of CPX-351 in children was established in the last year, and in March 2021, the FDA approved a revised label for CPX-351 to treat newly diagnosed t-AML or AML with MDS-related changes in pediatric patients aged 1 year and older.

Two different clinical trials, theCPX-MA-1201 trial (NCT01943682) and AAML 1421 (NCT02642965), explored the effectiveness of CPX-351 in pediatric AML. CPX-MA-1201, a phase I study of CPX-351 monotherapy, reported CR in two patients and one undetectable MRD among six treated pediatric patients with r/r AML. The AAML 1421 (NCT02642965), a phase I/II evaluating the effectiveness of CPX-351 combined with fludarabine, cytarabine, and granulocyte colony-stimulating factor(FLAG) in 39 pediatric patients in first relapse, reported CR in 20 (54%) patients, 5 CR with partial recovery of platelet count (CRp; 14%), and 5 CR with incomplete blood count recovery (14%) with no differences in terms of age-related toxicity profile for the drug [84].Furthermore, encouraging results were reported for the two-arm randomized CPX351-301 trial (NCT01696084) showing the superiority of CPX-351 versus 7 + 3 chemotherapy in term of the 5-year OS rate (18% in the CPX arm and 8% in the standard chemotherapy arm, respectively) in AML adult patients [85]. A phase III trial is currently enrolling pediatric patients up to 22 years old with newly diagnosed AML with or without FLT3 mutations, to compare standard chemotherapy to therapy with CPX-351 and/or gilteritinib (NCT04293562).

### 4.4. Tyrosine Kinase Inhibitors (i): c-KIT, Ras, FLT3i

C-KIT (CD117), a tyrosine kinase receptor expressed by hematopoietic progenitors [86], plays a critical role in cellular proliferation, differentiation, and apoptosis. Gain-of-function mutations result in constitutive c-KIT activation, which promotes the proliferation and resistance to apoptosis of hematopoietic stem cells [87]. In pediatric AML, kit mutations, particularly expressed by AML with CBF alterations, were associated with a higher cumulative incidence of relapse [88], although the relationship between kit mutations and worse outcome is controversial [89]. For this reason, the inhibition of c-kit could represent a good strategy against poor-risk pediatric AML, but currently, there are no specific inhibitors available for clinical study [90]. However, other kinase inhibitors showed activity against the c-kit receptor. Dasatinib, a BCR-ABL1 inhibitor, has off-target activity against multiple kinases, including c-kit [91]. The French Acute Myeloid Leukemia Intergroup analyzed the response to dasatinib in high-risk CBF AML patients(including high white blood cell count, KIT and/orFLT3mutations, and a less than 3-log MRD reduction) presenting less than 3-log MRD reduction before the second high-dose cytarabine (HiDAC) consolidation course or with molecular recurrence with no sibling available received dasatinib as a 1-year post-consolidation maintenance treatment. One-year DFS was 31.5%and two-year DFS was 25.7%.The median time to hematologic relapse was 6.0 months after dasatinib therapy initiation [92]. The CALG B study group showed a 67% 3-year DFS in previously untreated patients with c-KIT CBF AML [93].

A Chinese trial (NCT03173612) is currently assessing efficacy of dasatinibin CBF pediatric AML harboring c-kit mutation, but the results are still pending.

The Ras proteins H, K, and N are small GTPases involved in cellular signaling pathways, such as MAPK and PI3K/mTOR, that drive cell proliferation, differentiation, and growth. Ras may be a potential therapeutic target [89]. Compounds that target Ras/MAPK and PI3K/mTOR were tested in adult AML but, to date, no clinical trials have been reported in pediatric patients [94,95,96].

FMS-like tyrosine kinase 3 (FLT3) is a tyrosine kinase receptor expressed by immature hematopoietic cells but is lost during cellular differentiation and plays a critical role in stem cell proliferation, survival, and differentiation [97].

In pediatric AML, the incidence of internal tandem duplication(ITD) and tyrosine kinase domain (TKD) FLT3 mutations are similar, including the pediatric-specific FLT3-TKD mutations subgroup. FLT3-ITD mutations are associated with a dismal outcome [20]. While in adults the use of FLT3 inhibitors was shown to improve outcomes, clinical trials are still limited in pediatrics, and none of the inhibitors has been approved for use in this setting.

First-generation FLT3 inhibitors midostaurin, sorafenib, lestaurtinib, and sunitinib are promiscuous multi-kinase inhibitors that show activity not only against FLT3 but also against other kinases, such as KIT, platelet-derived growth factor receptor (PDGFR), and vascular endothelial growth factor (VEGF), leading to increased toxicity associated with off-target inhibition [98].

Midostaurin has been extensively tested in adult AML. Based on results of the RATIFY trial, it was approved in adults with de novo FLT3-mutated AML combined with induction chemotherapy [99]. To date, one phase I/II study of midostaurin monotherapy was reported in pediatric leukemias, but the trial was closed early due to lack of enrolments. This study showed limited efficacy in 22 patients with r/r FLT3-mutated AML and KMT21-r-ALL, with a median OS of 3.7 months and 1.4 months, respectively. Given the efficacy of midostaurin combined with chemotherapy in adult AML, the NCT03591510 trial is currently evaluating this combination in children with newly diagnosed FLT3-mutated AML [98]. Sorafenib is a multi-kinase inhibitor, which no prolonged EFS or OS when combined with standard chemotherapy for de novo FLT3-mutated adult AML [100,101].Promising activity of sorafenib was seen in pediatric AML. In fact, in children, the rate of sorafenib conversion into its active metabolite, sorafenib N-oxide, is higher than in adults, resulting in higher exposure and potentially greater efficacy [102,103]. The anti-leukemic activity of sorafenib was initially demonstrated in pediatric r/r AML patients enrolled in a phase I study conducted by St. Jude Children’s Research Hospital. Significant indication of efficacy was seen with both sorafenib monotherapy, which produced a reduction of blast counts, and sorafenib combined with clofarabine and cytarabine, demonstrating CR and partial responses (PRs) among treated patients, regardless of FLT3 mutations [103]. Similarly, in the concomitant phase I COG study, two children with r/r FLT3-mutated AML achieved clearance of blasts and proceeded to HSCT [104]. Moreover, encouraging retrospective data regarding sorafenib after pediatric HSCT, are available [105]. Finally, the phase III COG AML 1031 trial evaluated the efficacy of sorafenib in high-allelic-ratio FLT3-ITD AML. Sorafenib plus chemotherapy was compared to standard chemotherapy: CR rates following induction I and induction II were 73% vs. 56% (*p* = 0.078) and 91% vs. 70% (*p* = 0.007) in two subgroups, respectively. EFS rates at 3 years were 57.5% vs. 34.3% (*p* = 0.007), but no significant differences were observed in 3-year OS between the two treatment arms (63.9% vs. 54.1%, *p* =0.375), probably due to the initiation of TKI therapy after withdrawal from protocol therapy or at time of relapse [106].

Interestingly, lestaurtinib was evaluated in the AAML06P1 trial, but this phase I/II study was closed before efficacy could be assessed [107]. More potent and selective second-generation inhibitors were designed to reduce off-target activity and enhance FLT3 inhibition compared to first-generation FLT3 inhibitors. Quizartinib is a second-generation FLT3 inhibitor approved in Japan for the treatment of r/r FLT3-AML but has not been approved by other countries because of the lack of a significant benefit-to-risk ratio [108,109], although a recent clinical trial conducted in older patients unsuitable for intensive chemotherapy showed a median OS of 13.7 months in patients treated with quizartinib in combination with low-dose ARA C (LDAC) compared with 4.2 months in patients receiving LDAC alone, leading to a reconsideration of the role of quizartinib in FLT3-mutated AML [110]. Quizartinib combined with cytarabine and etoposide was evaluated in pediatric MLL-rearranged ALL and relapsed AML, regardless of FLT3-ITD status. Responses in the sevenFLT3-ITD AML cases were documented as follows: one CR, one CRp, one CR with incomplete neutrophil and platelet recovery, four stable diseases (SD) with marked lower blast counts [111].

In adults, gilteritinib, the newest third-generation FLT3 inhibitor, has become the new standard of care for management of r/r FLT3-mutated AML, based upon greater efficacy and less toxicity compared to standard chemotherapy alone [112].Two ongoing clinical trials are investigating this molecule in pediatric patients: the phase I/II study of gilteritinib plus chemotherapy with FLAG is currently open for children, adolescents, and young adults(6 months to <21 years of age) with r/r FLT3-mutated AML (NCT04240002).The other phase III randomized trial is evaluating the efficacy of standard chemotherapy (daunorubicin, cytarabine, and gemtuzumab ozogamicin) with or without gilteritinib vs. CPX-351 with or without gilteritinib in patients up to 22 years with newly diagnosed AML with or without FLT3 mutation, (NCT04293562), as mentioned above.

Crenolanib is a potent and selective inhibitor of FLT3, PDGFRα/β, and KIT, showing encouraging activity in adult AML. In pediatric FLT3-mutated hematologic malignancies, NCT02270788, a phase I trial, is currently open to evaluate feasibility combined with sorafenib.

### 4.5. Ruxolitinib: Is It a Possible Agent in AML?

Another group of molecular aberrations of increasing interest in myeloid malignancies are activating mutations of Janus kinases (JAK1, 2, and 3 genes) [113]. Ruxolitinib is a potent and selective JAK1 and JAK2 inhibitor, approved in adults for the treatment of intermediate- or high-risk myelofibrosis. The safety and tolerability of this agent in pediatric patients was documented in a phase ICOG trial. Children with refractory or relapsed malignancies (solid tumors, r/r ALL, r/r AML, and myeloproliferative neoplasms) have not shown any objective responses, but the study enrolled leukemias without JAK2 mutations [114]. Ruxolitinib was well tolerated also in heavily pretreated AML in a phase I/II clinical trial (NCT01251965); however, this trial closed because of the lack of clinical benefit. Although the study design was to enroll patients older than 14 years, the youngest patient was 25 years old [115].

Based on these results, future clinical studies will clarify the role of target agents combined with standard chemotherapy. One of these is currently ongoing and will evaluate the efficacy and feasibility of combination chemotherapy with target agents according to the result of targeted deep sequencing in pediatric patients with r/r solid tumor or AML (NCT02638428).

### 4.6. Ubiquitin–Proteasome Inhibitors

The proteasome is a large protein complex responsible for the degradation of most cellular proteins under physiological conditions, regulating cellular processes, such as cell survival and signaling [115]. One of molecular consequences of proteasome inhibition is the degradation of the NF-kB regulator IB resulting in the suppression of NF-kB activity and the accumulation of the two tumor-suppressor proteins p27 ^KIPI^ and p53 [116,117,118].Preclinical data from pediatric studies showed increased proteasome activity and NFkB levels in AML blasts [118,119]. Indeed, bortezomib, a first-generation proteasome inhibitor, approved for multiple myeloma and non-Hodgkin lymphoma, may be a useful approach to AML treatment.

Based on this, the Children’s Oncology Group developed the phase III AAML1031 trial to evaluate whether the addition of bortezomib to standard chemotherapy improved survival in pediatric patients with newly diagnosed AML [120]. All patients enrolled receive standard chemotherapy with or without bortezomib, according to randomized criteria, followed by HSCT for high-risk patients. No differences in EFS and OS were observed in the two study arms. Specifically, the 3-year EFS in the two arms with chemotherapy and bortezomib plus chemotherapy was 44.8% ± 4.5% vs. 47.0% ± 4.5% (*p* =0.236), and the 3-year OS was 63.6% ± 4.5% vs.67.2% ± 4.3% (*p* =0.356), respectively. In addition, subgroup analyses by risk group stratification showed similar outcomes for both low- and high-risk patients. In particular, the 3-yearDFS and OS for low-risk patients were 52.9% ± 3.7% and 74.1% ± 3.4%, respectively, while for high-risk AML they were 27.8% ± 6.6% and 36.9% ± 7.6%, respectively [120].

Neddylation involves post-translational enzymatic conjugation of an ubiquitin-like protein, NEDD8 (neural precursor cell-expressed, developmentally downregulated gene 8), to lysine residues on the target protein. Enzymatic cascades in this process involve NEDD8-activating enzyme E1 (NAE), NEDD8-conjuagating enzyme E2, and substrate-specific NEDD8-E3 ligases. This process regulates tumor-suppressor and oncoprotein activity and is dysregulated in several malignancies, including AML, making it an attractive therapeutic target [121].In particular, NAE regulates the activity of the cullin-RING E3 ubiquitin ligases, which controls the degradation of proteasome-regulated proteins with important roles in cell-cycle, DNA damage, stress responses, and signal transduction [122].

Pevonedistat (MLN4924) is a novel NAE inhibitor that blocks the first step of the neddylation cascade and has single-agent anti-proliferative activity in adult r/r AML [123].

There is an ongoing phase I trial (NCT03813147)to evaluate the side effects and efficacy of pevonedistat combined with azacytidine, fludarabine phosphate, and cytarabine in patients up to 21 years with r/r AML and MDS.

### 4.7. TP53 and MDM2 Antagonists

TP53 aberrations are rare in pediatric AML [124]. Alternative mechanisms to block wild-type p53 have been identified. In particular, over expression of MDM2 (murine double minute 2), a negative regulator of p53, induces p53 inactivation and degradation, resulting in the alteration of cell-cycle phases, DNA repair mechanism, and apoptosis. This mutation is expressed in up to 50% of AML [125,126].Therefore, there is significant interest in MDM2 antagonists as promising anticancer agents to block MDM2–p53 interactions and restore the tumor-suppressor functions of wild-type p53 [127,128].

Currently, a phase I study (NCT03654716) is evaluating the efficacy of the dual MDM2/MDMX inhibitor, ALRN-6924, as a possible treatment for refractory solid tumor, brain tumor, lymphoma, or leukemia (in particular, r/r AML, ALL, mixed-lineage leukemia, biphenotypic leukemia, or other undifferentiated acute leukemia) in pediatric patients and young adults (aged from 1 to 21 years old).

### 4.8. BCL-2 Inhibitors

Venetoclax is a highly selective oral inhibitor of the B-cell lymphoma-2 (BCL-2) protein, an anti-apoptotic member of the BCL-2 family of proteins also including BCL-XL, MCL1 (myeloid cell leukemia sequence 1). Overexpression of BCL-2 and other anti-apoptotic proteins has been documented in different type of malignancies, including AML [129,130].

Venetoclax was approved by both the FDA and the EMA combined with azacytidine, decitabine, or low-dose cytarabine for de novo AML in adults 75 years old or who have comorbidities precluding intensive induction chemotherapy, based on a CR rate of 70% [131].

To date, there are limited data on the efficacy of venetoclax in pediatric AML and MDS, but many trials are ongoing to evaluate the efficacy and possible toxicity in this population [132].

In the single-center experience of the Children’s Hospital of Colorado, venetoclax was administered in combination with azacytidine to pediatric r/r AML or high-risk MDS, showing good responses with three CR. Although a larger patient cohort is needed to confirm the safety and efficacy of venetoclax-based regimens for pediatric AML, this report demonstrated promising rates of remission induction, also in patients who are refractory to cytotoxic chemotherapy [133].

Moreover, Karol et al. reported exciting results of a phase I trial combining venetoclax with low- and high-intensity chemotherapy in 38 patients aged 2–22 years with r/r AML or acute leukemia of ambiguous lineage: a CR rate of 70% and overall response rate (ORR) of 80% was reported among 20 treated patients. As noted by the authors, this response rate is consistent with observations made by the two major studies in relapsed AML from the European and American pediatric oncology cooperative groups. Additionally, favorable outcomes with venetoclax were reported for genetic subtypes of pediatric AML including mutated *CBF*β/*MYH11*, *RUNX1-RUNX1T1*, and *CEBPA*, while no benefit was reported for *FLT3*-AML [134].

### 4.9. IDH and Menin Inhibition

IDH1/2 mutations occur in approximately 2–4% of pediatric AML, mostly in adolescents. Mutations in IDH1/2 impair cellular differentiation, generating high levels of 2-hydroxyglutarate that inhibit various components of the epigenetic machinery. Currently, NCT02813135 is ongoing to evaluate the activity of enasidenib as a single agent in10pediatric r/r IDH2-mutated AML cases. Moreover, a phase II trial, NCT04203316, is investigating the safety of enasidenib in pediatric r/r AML. No clinical trials with ivosidenib are open in pediatric AML, but NCT04195555 studies the efficacy of ivosidenib in treating pediatric patients with solid tumors or histiocytic disorders with IDH1 genetic mutations [135,136,137].

Menin inhibitors are novel, targeted agents currently in clinical development. Menin has a tumor-suppressor function in endocrine glands and is critical for leukemogenesis in a subgroup of mixed-lineage leukemia (MLL), driven by the rearrangement of the KMT2A gene, which encodes an epigenetic modifier. Menin inhibitors prevent the binding between MLL and menin, halting the oncogenic function of the KMT2A complex. The first trial in humans is NCT04065399, a phase I-II trial, which analyzes the efficacy and dose-limiting toxicities of a small molecule, SNDX-5613, in adults and children with r/r acute leukemia. Fifty-four patients recruited had a median of three prior therapies, and 45 patients were NMP1 or KMT2A mutated. The trial has demonstrated that the composite CR rate in the last subgroup of AML was 44% (20/45 patients) [138,139].

## 5. Conclusions

To date, the gold standard in the treatment of pediatric patients with AML is multiagent chemotherapy. For pediatric patients with high-risk features and those who relapse, allo-SCT is standard in first- and second-CR, respectively. Clinical trials are critical for advancing the treatment of pediatric patients with AML. Advances in cancer genomics and research in stem cell biology are providing new information about leukemogenesis and pharmacogenetics. Potential therapeutic approaches are advancing personalized treatment strategies, which may modify the clinical course of this pathology.

## Figures and Tables

**Table 1 biomedicines-10-01405-t001:** Characteristics of pediatric high-risk AML.

Genetic Risk Criteria	Response to Treatment Criteria
Complex karyotype (≥3 aberrations including at least one structural aberration)	MRD ≥ 1% after induction course 1 or ≥0.1% at induction 2 or blast count is ≥5% at induction 1
Monosomal karyotype, i.e., -7, -5/del(5q)	
11q23/KMT2A rearrangements involving: - t(4;11)(q21;q23) KMT1A/AFF1 - t(6;11)(q27;q23) KMT2A/AFDN - t(10;11)(p12;q23) KMT2A/MLLT10 - t(9;11)(p21;q23) KMT2A/MLLT3 with other cytogeneticaberrations	
t(16;21)(p11;q22) FUS/ERG	
t(9;22)(q34;q11.2) BCR/ABL1	
t(6;9)(p22;q34) DEK/NUP214	
t(7;12)(q36;p13) MNX1/ETV6	
Inv3(q21q26)/t(3;3)(q21;q26) RPN1/MECOM	
12p abnormalities	
FLT3-ITD with AR ≥0.5 not in combination with other recurrent abnormalities or NPM1 mutations	
WT1 mutation and FLT3-ITD	
inv(16)(p13q24) CBFA2T3/GLIS2	
t(5;11)(q35;p15.5) NUP98/NSD1 and t(11;12)(p15;p13) NUP98/KDM5A	
Pure erythroid leukemia	

**Table 2 biomedicines-10-01405-t002:** Target therapy for pediatric AML.

Therapeutic Mechanism	Pediatric AML Approval	Clinical Trials	Adult AML Approval	Comments
Hypomethylating Agents				
Azacytidine	Not approved	NCT02450877 NCT01861002 NCT03164057	Approved	These agents are incorporated into DNA resulting in downregulation of oncogenes, reactivation of tumor suppressors, and increasing sensitivity to cytotoxic agents.
Decitabine	Not approved	NCT01177540	Approved	
Histone Deacetylase Inhibitors				
Panobinostat	Not approved	NCT02676323	Not approved	HDAC inhibitors induce cell cycle arrest and apoptosis. In adult patients, panobinostat and vorinostat are approved in r/r multiple myeloma by the EMA and FDA and in advanced primary cutaneous T-cell lymphoma by the FDA, respectively.
Vorinostat	Not approved	NCT03263936	Not approved	
Pinometostat	Not approved	NCT02141828 NCT03724084	Not approved	
Immunotherapy and Immune-MediatedChemotherapy				
Gemtuzumabozogamicin (GO)	Approved	NCT00372593	Approved	The FDA approved GO for - Newly diagnosed CD33-positive AML in adults. - r/r CD33-positive AML in adults and in pediatric patients 2 years and older. The EMA approved GO for patients aged 15 years (AYA) and above who are newly diagnosed and have not tried other treatments. It is used in combination with daunorubicin and cytarabine.
CD123-targeting drugconjugate	Not approved	NCT02848248 NCT03386513	Not approved	These therapiesremain in the early phases of research.
Ipilimumab	Not approved	NCT00039091 NCT02890329 NCT00060372	Not approved	Immune check point inhibitors have shown a good clinical response in combination with other drugs. These trials recruited patients up to 18 years old; NCT03825367 is for pediatric r/r AML. Ipilimumab is approved in solid tumors andpembrolizumab and nivolumab in Hodgkin’s lymphoma and solid tumors by both FDA and EMA.
Pembrolizumab	Not approved	NCT03291353 NCT02996474 NCT02845297 NCT02768792 NCT02771197 NCT03286114 NCT02981914	Not approved	
Nivolumab	Not approved	NCT02397720 NCT02532231 NCT02275533 NCT02846376 NCT03825367	Not approved	
Car-T Cells				
Anti-CD33	Not approved	NCT03971799 NCT03927261 NCT03126864	Not approved	Early phase trials are planned including phase I/II trials in children and young adults.
Anti-CD123	Not approved	NCT02159495 NCT04318678	Not approved	
Tyrosin Kinase Inhibitors				
Dasatinib	Not approved	NCT03173612	Not approved	Results from pediatric AML trials are ongoing. Dasatinib is approved in both adult and pediatric hematological malignancies.
Midostaurina	Not approved	NCT00866281 NCT03591510	Approved	It is approved in newly diagnosed FLT3+ AML adult patients.
Sorafenib	Not approved	NCT01518413 NCT00908167 NCT00665990 NCT01371981 NCT01445080	Not approved	Clinical trials are ongoing in both pediatric and adult patients.
Quizartinib	Not approved	NCT01411267	Not approved	-
Listaurtinib	Not approved	NCT00469859 NCT00557193	Not approved	-
Gilteritinib	Not approved	NCT04240002 NCT04293562	Approved	It is approved in adult r/r AML with an FLT 3 mutation by the FDA and EMA.
Crenolanib	Not approved	NCT02270788	Not approved	-
JAK Inhibitors/Ruxolitinib				
Ruxolitinib	Not approved	NCT01251965 NCT02638428	Not approved	It is approved in adults for intermediate and high-risk myelofibrosis.
Proteasome/Ubiquitin/NEDD8 Inhibitor				
Bortezomib	Not approved	NCT01371981	Not approved	It is approved for multiple myeloma and non- Hodgkin lymphoma.
Pevonedistat	Not approved	NCT03813147	Not approved	-
TP53/MDM2 Antagonists				
TP53/MDM2 antagonists	Not approved	NCT03644716	Not approved	-
BCL2 Inhibitors				
Venetoclax	Not approved	NCT03236857 NCT03194932	Approved	Approved for CLL and AML in adults 75 years old or unfit.
IDH Inhibitors				
Enasidenib	Not approved	NCT02813135	Approved	Approved for the treatment of adult patients with r/r AML with IDH2 mutation.
Ivosidenib	Not approved	Not open	Approved	Approved in adult R/R AML IDH1-mutated (or first line in elderly patients with AML). NCT04195555 is currently ongoing to investigate ivosidenib in pediatric solid tumors and lymphomas with IDH1 mutations.
Menin Inhibitors				
Sndx-5613	Not approved	NCT04065399	Not approved	Multiple clinical trials are ongoing. Preliminary results in r/r*NPM1*mutant and*KMT2Ar*AML have shown tolerable toxicity and promising clinical activity (see below in the text).

## Data Availability

Not applicable.

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
