# Peer review of "High-Risk Acute Myeloid Leukemia: A Pediatric Prospective"

_biomedicines, 2022, doi:10.3390/biomedicines10061405_

Round 1

Reviewer 1 Report

F Cacace and colleagues submit an Interesting and extensive review of all the therapeutical possibilities in High risk Acute Myeloid Leukemia in Pediatric patients.

More informations are available in the Adult population, but some sub-types cross, others don't. Difficult to establish conclusions from adults data for pediatrics patients.

After the list of all the possibilities it can be of interest to submit recommandations in the context of high risk AML in children and AYA, especially in the context of Relapse/refractory situations.

Derived from adults data, for example,  it will difficult in the context of FLT3 mutated AML to put at the same level the use of FLT3 inhibitors and the use of Immunotherapy (CAR-T or Bispecific Antibody).

Maybe, a simple thing will be to put in the table 1 what are the drugs already approved in the context of adults patients.

Reviewer 2 Report

The authors have written a perspective on pediatric high-risk AML. I am not an English-native person, but probably it should either be 'pediatric perspective' or 'pediatric prospect'.

Some comments:

Main:

  • introduction: the biggest challenge is to avoid relapse, not so much a higher CR rate or improving OS.
  • introducion: new treatment approaches should ultimately lead to less relapses, not to more options in case of relapse.
  • pathophysiology: I do not understand why there would be an increased incidence of AML with underlying disorders such as Fanconi etc.; please explain
  • pathophysiology: this manuscript is (focussed) on pediatric AML, so the percentages given on FLT3 mutations etc. must reflect the pediatric data. Alternatively, explain these are the incidences in adults and add the pediatric data
  • Table: GO is not approved for children of all ages by EMA, please add it is approved for children aged 15 years and above and by the FDA also in younger children. In addition, menin inhibitors and IDH1/2 inhibitors must be mentioned
  • Illustrations: the manuscript would benefit from more illustrations

Minor:

  • the abstract states that prognosis is poor, however, most recent studies show up to 80% overall survival. Arbitrarily, this is not poor. While the introduction is somewhat more nuanced, a review from 2011 is too old and certainly literatire shows 3-year overall survival rates above 70%
  • section 4.2 on immunotherapy: when mentioning outcome in the patients that received GO, please mention the outcome in the other group not getting GO, so that the readers can  understand the actual impact of GO 
  • COG very recently published on sorafenib in JCO, please add that reference (data are included, but do not seem to have a reference)
  • Quizartinib was effective in a trial in adults, although data have not been published yet to the best of my knowledge
  • when discussing the results on venetoclax, experience from European and American pediatric oncology cooperative groups is mentioned, but not supported by 1 or 2 references or a review on this

Author Response

For a mistake, we are not able to upload the answer to reviewer 1, which find below.

Reviewer 3 Report

Cacace et al submitted a manuscript on pediatric AML, describing several novel therapies that are being studied. The paper is summing up several of these agents but is a bit flat as there is no clear direction which agents are most promising and urgently need to be studied in the opinion of the authors. Moreover some details need revision as they are not entirely reflecting the current ped AML landscape, as detailed below.

Comments/suggestions:

  • Abstract: ped AML as a whole has nowadays OS rates in the 70-80% range, that is no longer “Poor outcome”
  • The title suggests ‘high risk AML’ but the authors do not provide a sufficient explanation how they define this
  • Introduction line 38: the 15% OS is for second relapse not first relapse, OS rates in 1st relapse are in the 40-50% range, see recent BFM SG papers in Cancer and the pivotal paper from Kaspers et al about the 2001/01 study.
  • Line 77: I do not think one finds 27% ped AML cases to be NPM1 mutated
  • Line 75: CBF also includes inv(16)
  • Line 89: Ped AML in fact has more fusion gene abnormalities and less single gene mutations that adults, but these are certainly present in the WHO classification
  • Line 109: I do not think that treatment in ped AML is driven by adult AML – in general we treated with more chemotherapy and less SCT – and therapy for elderly AML is completely different
  • Table 1: check point inhibitors are not studied intensively in ped AML, and studies in adults have not been extremely encouraging, not surprising as ped AML often lacks neoantigens as it is genomically stable, this needs to be better addressed and I would not present table 1 as if an array of studies is ongoing
  • The section on epigenetic therapies ends with DOT1L but all interest for MLL is now shifted to menin inhibitors which are not mentioned
  • Mylotarg is discussed as immunotherapy but in fact is targeted delivery of chemotherapy and not immunotherapy
  • Mylotarg: I would reference the original studies that led to the re-approval of Mylotarg for adults as fractionated therapy rather than high-dose.
  • In the Vyxeos part it is important to discuss the lack of availability of daunoxome, and whether vyxeos can fill that gap – related to the issue of cardiac toxicity
  • Dasatinib has been studied in adult CBF AML by the French group – please add these data
  • Line 356: I do not think there is convincing evidence that FLT3 KD mutations confer poor outcome
  • What is the hypothesis that sorafenib did not translate in OS differences in the COG study ?
  • A quizartinib adult trial was presented at ASH and reached its primary endpoint – hence there is now favorable benefit-risk ratio
  • The article has various English spelling errors which may need to be corrected
